# Health system's availability and readiness of health facilities for chronic non-communicable diseases: Evidence from the Ethiopian national surveys

**Atkure Defar[1,2], Girum Taye Zeleke[1], Della Berhanu[1,3], Ephrem Tekle Lemango[4], Abebe Bekele[1], Kassahun Alemu[2], Sibhatu Biadgilign** [5]*

1 Ethiopian Public Health Institute, Addis Ababa, Ethiopia, 2 College of Medicine and Health Science, Institute of Public Health, Department of Epidemiology and Biostatistics, University of Gondar, Gondar, Ethiopia, 3 London School of Hygiene & Tropical Medicine, London, United Kingdom, 4 Maternal, Child Health and Nutrition Directorate, Ministry of Health, Addis Ababa, Ethiopia, 5 Independent Public Health Analyst and Research Consultant, Addis Ababa, Ethiopia

* sibhatu2010@gmail.com

**Data Availability Statement:** The data belong to the National Data Management Center (NDMC) at the Ethiopian Public Health Institute (EPHI).

## Abstract

### Introduction

Non-communicable diseases (NCDs) currently cause more deaths than all other causes of deaths. Cardiovascular disease, diabetes, cancer, and chronic respiratory diseases-threaten the health and economies of individuals and populations worldwide. This study aimed to assess the availability and readiness of health facilities for chronic non-communicable diseases (NCDs) and describe the changes of service availability for common NCDs in Ethiopia.

### Methods

We used data from the 2014 Ethiopia Service Provision Assessment Plus (ESPA +) and 2016 and 2018 Service Availability and Readiness Assessment (SARA) surveys, which were cross-sectional health facility-based studies. A total of 873 health facilities in 2014, 547 in 2016, 632 in 2018 were included in the analysis. (ESPA+) and SARA surveys are conducted as a census or a nationally/sub-nationally representative sample of health facilities. Proportion of facilities that offered the service for diabetes, cardiovascular disease, chronic respiratory disease, cancer diseases, mental illness, and chronic renal diseases was calculated to measure health service availability. The health facility service readiness was measured using the mean availably of tracer items that are required to offer the service. Thus, 13 tracer items for diabetes disease, 12 for cardiovascular disease, 11 for chronic respiratory disease and 11 cervical cancer services were used.

### Results

The services available for diagnosis and management did not show improvement between 2014, 2016 and 2018 for diabetes (59%, 22% and 36%); for cardiovascular diseases (73%, 41% and 49%); chronic respiratory diseases (76%, 45% and 53%). Similarly, at the national

National health and health related data have their own data governance system and shall be managing under the national data sharing policy. Anyone interested to use the data sets could communicate to the Ethiopian Public Health institute, National Data Management Centrer using the following link: https://rtds.ephi.gov.et/public/showdetail/94#step-1. The data can be accessed using following URLs: ESPA+ 2014 (https://rtds.ephi.gov.et/public/showdetail/76), SARA 2016 (https://rtds.ephi.gov.et/public/showdetail/93) and SARA 2018 (https://rtds.ephi.gov.et/public/showdetail/94).

**Funding:** The authors received no specific funding for this work.

**Competing interests:** The authors have declared that no competing interests exist.

level, the mean availability of tracer items between 2014, 2016 and 2018 for diabetes (37%, 53% and 48%); cardiovascular diseases (36%, 41% and 42%); chronic respiratory diseases (26%, 27% and 27%); and cancer diseases (6%, 72% and 51%). However, in 2014 survey year, the mean availability of tracer items was 7% each for mental illness and chronic renal diseases, respectively.

## Conclusions

The majority of the health facilities have low and gradual decrement in the availability to provide NCDs services in Ethiopia. There is a need to increase NCD service availability and readiness at primary hospitals and health centers, and private and rural health facilities where majority of the population need the services.

## Introduction

According to the World Health Organization (WHO) each year the deaths of 40 million people were reported from non-communicable diseases (NCDs), which is equivalent to 70% of all deaths globally [1]. In Ethiopia, NCDs cause 42% of deaths, of which 27% are premature deaths before 70 years of age [2]. The Disability Adjusted Life Years (DALYs) increased from below 20% in 1990 to 69% in 2015 [3] and there is a rapidly increasing burden of NCDs in Ethiopia [2, 4–9]. Furthermore, while the annual number of deaths due to infectious diseases is projected to decline, the total annual number of NCD deaths is projected to increase to 52 million by 2030 [10–12].

The main types of NCDs are cardiovascular diseases (such as heart attack and stroke), cancers, chronic respiratory diseases (CRDs) (such as chronic obstructive pulmonary disease and asthma) and diabetes [1]. Cardiovascular diseases (CVDs) account for most of the annual NCD deaths (17.7 million), followed by cancers (8.8 million), respiratory diseases (3.9 million), and diabetes (1.6 million) [1]. A study of the prevalence of raised blood glucose and diabetes in Ethiopia was 6% and raised blood pressure (SBP > 140 and/or DBP > 90 mmHg) was 15.6%, [13].The overall death rates of NCDs causes has shown a decline by 40% from 1990 to 2015. However, the crude all-cause mortality rate in 2015 was 680.9 per 100,000 people (95% UI: 505.1–913.9), of which 286.9 (95% UI: 188.1–423.0) per 100,000 were due to NCDs [14].

NCDs threaten the progress towards the 2030 Agenda for Sustainable Development, which includes a target of reducing NCD related premature deaths by one-third. In order to attain this target, each country should prioritize and implement cost-effective policies and interventions and provide health services which tackle the rapid surges of this silent epidemic. In low and middle income countries (LMICs), several interacting factors have been associated with high burden of NCDs, including few preventative and early detection services, low availability of essential treatments, and suboptimal organization in the delivery of care [15]. The assessment of the capacity of countries' to prevent and control NCDs conducted by WHO in 2015 also showed major gaps in health system capacity in many LMICs [16]. The absence of evidence-based policies and program with inadequate response at primary care level perpetuates the problem. With limited data from LMICs on NCD service readiness, availability and utilization in primary care facilities, the situation will become worse with time unless the situation is actively addressed at a critical time [17].

In Ethiopia, there are a few studies conducted on service availability and readiness for NCDs. These studies, however, are limited as they focus on a particular or selected set of NCDs [18–20]. In the country, the health facilities lack essential logistics and have major service provision bottlenecks including non-functionality of the existing district coordination body and technical committees, training gaps, staff shortage, high staff turnover resulting from work related burden, fatigue and poor motivation among service providers [21]. The overall aim of the current study was to assess the availability and readiness of services for common NCDs using Ethiopian Service Provision Assessment (ESPA+) and Service Availability and Readiness Assessment (SARA) surveys in health facilities of Ethiopia.

## Methods

### Data source and study setting

This study used and analyzed data from the 2014 Ethiopia Service Provision Assessment Plus (ESPA+), and 2016 and 2018 Service Availability and Readiness Assessment (SARA) surveys. (ESPA+) and SARA surveys are conducted as census or as a nationally/sub-nationally representative sample of health facilities that can guide the health sector policy and planning. The surveys were conducted using similar survey methodologies for adapting the questionnaires, training, and pre-testing and data collection. ESPA+ and SARA questionnaires contain information on general and specific service availability and readiness (e.g., types of services offered, basic amenities, equipment, medicines and commodities, diagnostics, trained staff and practice guidelines). The data were collected from all nine regions of Ethiopia namely: Tigray, Afar, Amhara, Oromia, Somali, Benishangul-gumuz, Southern Nations Nationalities and Peopple (SNNP), Gambella, and Harari. The survey also included the two city administrations, Addis Ababa and Dire Dawa. The Ethiopian health system has primary, secondary and tertiary levels of care. Five health posts, their referral health center and primary hospital comprise the primary health care unit [22].

### Study design

A cross-sectional health facilities-based study was employed in the surveys. The surveys used the SARA methodology that was developed through a joint WHO and the United States Agency for International Development (USAID) collaboration. The method considers, among other thing, the service availability mapping (SAM) tool developed by WHO, and the ESPA + tool developed by ICF International under the USAID-funded MEASURE DHS (monitoring and evaluation to assess and use results, demographic and health surveys) project [23].

### Sampling of health facilities

For ESPA+ 2014, the survey used a stratified random sampling designed to provide representative samples of: health facility types, management authorities, regions, and city administrations. A list of health facilities obtained from each regional health office was used as a sampling frame. This sampling method yielded a total of 1,327 health facilities to be surveyed, which comprised of 223 hospitals, 298 health centers and 321 health posts and 485 private clinics. In 2014 sampling, all hospitals were included and a representative sample of health centers, health posts and clinics were included. While, in 2016 sampling, all hospitals were included and a representative sample of health centers, health posts and clinics were included and in 2018 sampling, all hospitals were included and a representative sample of health centers, health posts and clinics were included. Briefly, the sampling method for SARA 2016 and 2018 was also a nationally representative sample stratified by health facility type and managing authority. All

health facilities that were in 2016 SARA were selected for the SARA 2018 survey. In the 2016 SARA survey, 705 health facilities were included, with all hospitals (228), 165 health centers, 173 clinics, and 139 health posts. Similarly, in the 2018 SARA survey, a total of 764 health facilities were assessed, which included 303 hospitals, 164 health centers, 165 clinics and 132 health posts. The 2018 SARA survey also included new hospitals that were established after the 2016 SARA survey [23–25].

## Data collection tool and procedure

An adapted version of the MEASURE DHS project SPA questionnaire with some additional country specific questions and World Bank service delivery indicator (SDI) were used to collect data on facility characteristics, staff contingents, service availability, referral systems, drugs and equipment, laboratory services, reporting systems, and monitoring and supervision [26]. To obtain this information, we interviewed heads of facilities or their delegates or heads of specific sections that were believed to be knowledgeable about the service provision. Inventory and inspection of the guidelines, reports, equipment, drugs and laboratory supplies were also done.

The survey data were collected using Computer-Assisted Personal Interviewing (CAPI) programed using the Census and Survey Processing System (CSPro) 6.3 software. Data were transferred daily to the Ethiopian Public Health Institute server. Feedback was provided to the field team before they moved to the next selected health facility. The Ethiopian Public Health Institute (EPHI) organized the fieldwork, monitored the data transfer system and carried out extensive supervision over the duration of the fieldwork. Regional coordinators engaged in supervision and actively solved problems faced by the field team.

## Data management

The survey data were collected focusing on service readiness and availability for all NCDs in the ESPA+, including diabetes, cardiovascular diseases, chronic respiratory diseases and cancer. To calculate the readiness score, 13 tracer items for diabetes, 12 for cardiovascular diseases, 11 for chronic respiratory diseases and 11 cervical cancer services were used. The availability of the services and readiness of health facilities for diabetes, cardiovascular, and chronic respiratory diseases were investigated in the ESPA+ 2014, and the two recent (2016 and 2018) SARA surveys [23–25]. The service was considered to be available if a facility manages, diagnoses, treats or prescribes a patient coming to the facility for common NCDs. The facility readiness was estimated based on the availability of tracer items that are necessary to provide the services [27].

## Measurement of variables

The outcome variables in this study were the "availability score" and "readiness score" of the health facilities for common NCDs. We defined facility ''availability" as the percentage of facilities in the sample that said they offered the service in question, which considered whether the diagnosis and/or management services were available. Facility "readiness" was a composite measure and was restricted to the subset of facilities that offered the service. The component "domains" that make up the readiness score included trained staff, guideline, equipment/supplies; medicines and commodities, and diagnostics capacities. For each domain, we calculated an index as the mean score of items expressed as a percentage. A readiness score of 50 signifies that, on average, half of the facilities that offered the service had each of the requisite inputs for delivering that service. The calculated indices were compared to an agreed cutoff point of 50%. A facility index that was below 50% was considered as 'not ready' to manage NCDs.

## Statistical analysis

Data were cleaned and analyzed using Stata 15 (StataCorp, College Station, TX). A descriptive analysis was carried out to summarize results using percentages and proportions, which were then presented in the form of tables and graphs. Percentages in the tables were weighted to make sampling adjustment for disproportional sampling and non-response rate at regional level and by facility type, but the sample size presented in the tables are not weighted. This is a guiding principle for the analysis of SPA data. The analysis for availability and readiness has been done considering the WHO service availably and readiness estimation method [1]. In this study, Fisher's exact test was used to assess the difference in the mean availability of tracer items for various NCDs and across survey years. Further statistical analyses were done by using chi-squared tests. Statistical significance was accepted at the 5% level (p<0.05). The analysis did not include health posts since, they do not have standards for NCDs, nor are they supposed to provide curative services for NCDs. The 'Strengthening the Reporting of Observational Studies in Epidemiology (STROBE) Statement' to write the manuscript (online S1 Checklist) was followed.

## Quality assurance

Quality assurance involved multiple steps along a continuum of training, data collection in the field, and data processing at the central level. Quality assurance began with the recruitment of data collectors and team leaders with a health background. Team leaders played a critical role in the correct completion of the data collection tools as they reviewed all the questionnaires. The regional coordinators visited and communicated with teams regularly to provide support and help when difficulties arose at individual facilities. The central coordinators at EPHI and members of the Technical Working Group supervised the teams in the field to ensure consistency and quality. When needed, the central data managers telephoned facilities for clarification and to ensure quality.

## Ethics approval and consideration

The current study was conducted based on a secondary analysis of existing data sets from the three-waves of data collection (2014, 2016, and 2018). Ethical clearance was obtained from the Scientific Ethical Review Committee (SERC) of the Ethiopian Public Health Institute (EPHI) with reference number of EPHI 6.13/966. Informed verbal consent was obtained from all health facilities' in-charge and health workers responsible for client services who were present at the facility. The participant recruitment date for SPA 2014 was from March to July, 2014; for SARA 2016 from February to April 2016 and for SARA 2018 from October to December, 2017.

# Result

## Characteristics of health facilities

In this comprehensive study, a total of 873 health facilities in 2014, 547 in 2016, and 632 in 2018 were included in the analysis. The results are stratified by facility type, managing authority, region, and residence (**Table 1**).

## Availability of important components for the management of NCDs

The services available for the diagnosis and/or management of diabetes did not show improvement between the periods of 2014 to 2018. The availability for diagnosis and/or management services for diabetes was 59%, 22%, and 36% in 2014, 2016, and 2018, respectively. Similarly,

**Table 1. Weighted percent of health facilities offering services for non-communicable diseases, by background characteristics, Ethiopia.**

| Background Characteristics | Diabetes | | | Cardiovascular diseases | | | Chronic respiratory diseases | | | Cancer | | | Number of surveyed facilities(unweighted)* | | |
|---|---|---|---|---|---|---|---|---|---|---|---|---|---|---|---|
| | | | | | | | | | | Cancer diseases | Cervical Cancer | | | | |
| | SPA + 2014 | SARA 2016 | SARA 2018 | SPA + 2014 | SARA 2016 | SARA 2018 | SPA + 2014 | SARA 2016 | SARA 2018 | SPA+ 2014 | SARA 2016 | SARA 2018 | SPA + 2014 | SARA 2016 | SARA 2018 |
| **Facility type** | | | | | | | | | | | | | | | |
| Referral Hospital | 97 | 91 | 97 | 97 | 84 | 97 | 97 | 91 | 97 | 69 | 56 | 77 | 31 | 32 | 31 |
| General Hospital | 97 | 90 | 97 | 95 | 92 | 97 | 96 | 92 | 95 | 66 | 38 | 64 | 132 | 117 | 116 |
| Primary Hospital | 98 | 89 | 95 | 100 | 89 | 94 | 100 | 89 | 96 | 65 | 10 | 42 | 48 | 61 | 156 |
| Health Center | 63 | 16 | 37 | 82 | 50 | 66 | 88 | 55 | 70 | 28 | 2 | 6 | 291 | 165 | 164 |
| Higher clinic | 89 | 69 | 48 | 87 | 61 | 48 | 88 | 69 | 48 | 60 | 4 | 29 | 58 | 23 | 19 |
| Medium clinic | 96 | 65 | 81 | 93 | 71 | 79 | 93 | 76 | 82 | 32 | 1 | 24 | 135 | 64 | 74 |
| Lower Clinic | 34 | 5 | 8 | 49 | 10 | 5 | 49 | 11 | 13 | 3 | 0 | 0 | 178 | 85 | 72 |
| **Managing authority** | | | | | | | | | | | | | | | |
| Government/ Public | 64 | 19 | 41 | 83 | 51 | 68 | 88 | 56 | 71 | 29 | 3 | 8 | 441 | 320 | 410 |
| Others** | 228 | 26 | 32 | 223 | 30 | 29 | 208 | 33 | 35 | 40 | 1 | 10 | 432 | 227 | 222 |
| **Location of health facilities** | | | | | | | | | | | | | | | |
| Urban | 72 | 34 | 43 | 78 | 43 | 53 | 82 | 45 | 57 | 28 | 4 | 14 | 536 | 431 | 494 |
| Rural | 49 | 5 | 25 | 70 | 38 | 41 | 72 | 45 | 47 | 19 | 0 | 1 | 337 | 116 | 138 |
| **Total** | 59 | 22 | 36 | 73 | 41 | 49 | 76 | 45 | 53 | 23 | 2 | 9 | 873 | 547 | 632 |

*considering the numbers of health facilities of each type and national and regional levels are weighted/ adjusted, so that the distribution of facilities by type's contribution to the total is proportionate to reflect the actual distribution or total number of health facilities in Ethiopia. We use weighted, for the percentages and unweighted for the sample size estimation.

**Others: Governmental (military, prison, federal police), private for-profit and NGO (mission/faith-based, nonprofit)

SPA: Service Provision Assessment; SARA: Service Availability and Readiness Assessment

the availability for diagnosis and/or management services for cardiovascular disease was 73%, 41%, 49% in 2014, 2016, and 2018, respectively and 76% in 2014, 45% in 2016 and 53% in 2018 for chronic respiratory disease. Mostly, the availability of essential guidelines, trained human power and materials and equipment for the management of NCDs services varied across the survey years. Detail description is illustrated in **Table 1**.

## Health facility readiness for the management of NCDs

Service readiness was measured generally by the availability of selected tracer items. In all categories of diseases, the readiness was checked by the availability of guideline and at least 1 trained staff as necessary tracer items to provide the service. Additionally, the analysis considered the availability items listed in **Table 2**. The general service readiness for cancer diseases, mental illness, and chronic renal diseases were measured based on the availability of guidelines and at least 1 trained staff as a common tracer item.

Service readiness was estimated among those who offered the services by assessing four domains: staff and guidelines, equipment, diagnostics, and medicines and commodities. The bar graph shows the trend on overall readiness using the availability of tracer items for three of the selected NCDs (diabetes, cardiovascular, cervical cancer and chronic respiratory diseases), which showed an improvement across different periods. We used 12 tracer items that are

**Table 2. Percentage of health facilities having guideline, trained staff, and equipment, diagnostic capacity and essential medicines for non-communicable diseases by year, Ethiopia.**

| Indicators by disease | SPA+2014 | SARA 2016 | SARA 2018 |
|---|---|---|---|
| **Diabetes** | | | |
| Guideline** | 12 | 16 | 15 |
| At least 1 trained staff | 6 | 10 | 8 |
| Blood Pressure apparatus | 93 | 100 | 96 |
| Adult weighing scale | 76 | 97 | 93 |
| Height board/stadiometer | 49 | - | - |
| Blood glucose | 40 | 66 | 66 |
| Urine protein | 56 | 89 | 78 |
| Urine glucose | 52 | - | - |
| Metformin | 11 | 31 | 20 |
| Glibenclamide | 28 | 31 | 22 |
| Injectable insulin | 9 | 18 | 17 |
| Injectable glucose | 15 | 62 | 57 |
| Measuring Tape | - | 71 | 69 |
| Urine dipstick ketones | - | 89 | 78 |
| Gliclazide/glipizide tablet | - | 4 | 7 |
| **Mean availability of tracer items** | **37** | **53** | **48** |
| **Cardiovascular disease** | | | |
| Guideline | 11 | 9 | 11 |
| At least 1 trained staff | 8 | 7 | 7 |
| Stethoscope | 97 | 100 | 97 |
| B. Pressure apparatus | 92 | 100 | 95 |
| Adult scale | 74 | 88 | 90 |
| ACE inhibitors (Enalapril) | 15 | 25 | 34 |
| Thiazide | 25 | 20 | - |
| Beta-blockers (Atenolol) | 6 | 15 | 17 |
| Calcium channel blockers | 26 | 20 | 30 |
| Oxygen | 9 | 17 | 13 |
| Hydrochlorothiazide tablet | - | 37 | 47 |
| Metformin | - | 19 | 16 |
| Asprin | - | 53 | 42 |
| **Mean availability of tracer items** | **36** | **41** | **42** |
| **Chronic respiratory disease** | | | |
| Guideline | 15 | 13 | 10 |
| At least 1 trained staff | 9 | 8 | 6 |
| Stethoscope | 99 | 100 | 96 |
| Peak flow meter | 1 | 5 | 4 |
| Spacers for inhalers | 1 | 4 | 6 |
| Salbutamol inhaler or tablets | 43 | 23 | 39 |
| Beclomethasone inhaler | 1 | 3 | 6 |
| Prednisolone | 44 | 44 | 47 |
| Hydrocortisone injection | 14 | 24 | 30 |
| Injectable epinephrine | 51 | 54 | 39 |
| Oxygen | 9 | 15 | 13 |
| **Mean availability of tracer items** | **26** | **27** | **27** |
| **Cancer diseases** | | | |

*(Continued)*

**Table 2.** (Continued)

| Indicators by disease | SPA+2014 | SARA 2016 | SARA 2018 |
|---|---|---|---|
| Guideline | 7 | - | - |
| At least 1 trained staff | 4 | - | - |
| **Mean availability of tracer items** | **6** | - | - |
| **Mental illness** | | | |
| Guideline | 9 | - | - |
| At least 1 trained staff | 5 | - | - |
| **Mean availability of tracer items** | **7** | - | - |
| **Chronic renal diseases** | | | |
| Guideline | 11 | - | - |
| At least 1 trained staff | 2 | - | - |
| **Mean availability of tracer items** | **7** | - | - |
| **Cervical cancer** | | | |
| Guideline | - | 53 | 49 |
| At least 1 trained staff | - | 61 | 30 |
| Speculum | - | 96 | 85 |
| Acetic acid | - | 77 | 38 |
| **Mean availability of tracer items** | - | **72** | **51** |

**Guideline is referring to a national guideline that the health worker should consider while treating patients for each particular diseases according to the standard set by the federal ministry of health

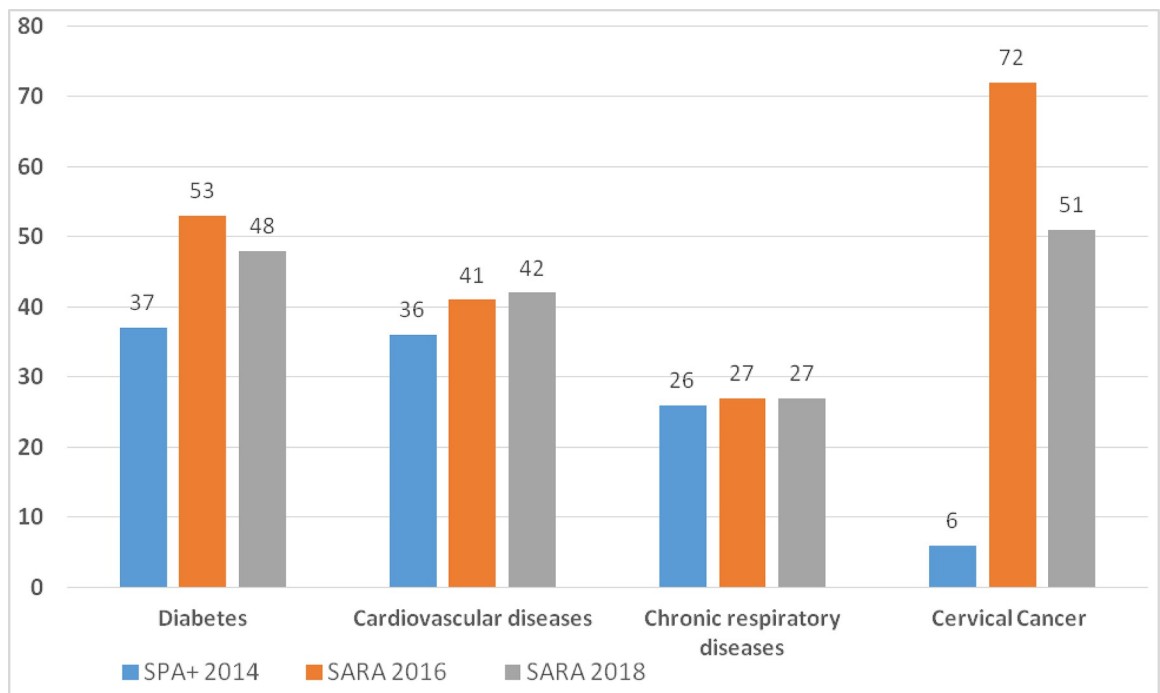

**Fig 1. Mean availability of tracer items among facilities that offer diabetes, cardiovascular diseases and chronic respiratory disease services; Ethiopia, SPA+ 2014, SARA 2016 and SARA 2018.**

**Table 3. Health facilities mean availability of tracer items difference in percent for non-communicable diseases service between assessment years at national level, Ethiopia.**

| Non-communicable diseases | SPA+2014 to SARA 2016 | | SARA 2016 to 2018 | |
|---|---|---|---|---|
| | Difference | P-value | Difference | P-value |
| Diabetes | 16 | < **0.001** | -5 | 0.07 |
| Cardiovascular disease | 5 | 0.12 | 1 | 0.72 |
| Chronic respiratory disease | 1 | 0.73 | 0 | 1.000 |
| Cervical Cancer | * | * | -21 | < **0.001** |

* Not applicable

necessary to provide diabetes services in 2014, whereas 13 tracer items were used for both 2016 and 2018 surveys. On average, for each survey years, facilities had 4, 6, and 6 tracer items with overall readiness score of 37%, 53%, and 48%, respectively. This showed improvement over the period between 2014 and 2016. The overall readiness for cardiovascular diseases (of the total 12) did not show improvement over time; it was 36%, 41%, and 42% during, 2014 (10 tracer items), 2016 (of 12 tracer items), and 2018 (12 tracer items), respectively (**Fig 1**).

Of the 11 tracer items that are necessary to provide diabetes service, on average facilities had 3 tracer items with an overall readiness score of 26%. There were no changes over time 2016 (27%) and 2018 (27%), with the mean tracer item availability of 3 out of 11 items. At the national level, the mean availability of tracer items for diabetes, cardiovascular diseases, chronic respiratory diseases, cancer diseases, mental illness, and chronic renal diseases were 37%, 36%, 26%, 6%, 7% and 7%, respectively in 2014. In 2016, the mean availability of tracer items for diabetes, cardiovascular diseases, chronic respiratory diseases and cervical cancer were 53%, 41%, 27% and 72%, respectively. Similarly, the mean availability of tracer items for diabetes, cardiovascular diseases, chronic respiratory diseases, and cervical cancer were 48%, 42%, 27% and 51%, respectively in 2018 (**Table 2**).

### Changes in preparedness for common non-communicable diseases

Table 3 shows the differences in the mean availability of tracer item in percent for NCDs service availability between assessment years at the national level. In this study, there was an improvement from 2014 to 2016 in terms of the availability of tracer items with 16% for diabetes (p< 0.001), 5% cardiovascular diseases, and 1% chronic respiratory diseases. However, from 2016 to 2018 survey year, there was a decrease for diabetes with 5% and 21% for cervical cancer (p< 0.001) and showed almost no change for cardiovascular and chronic respiratory diseases (**See Table 3**). The change in mental health and chronic renal diseases service readiness were not assessed since we have only one-time data for the 2014.

### Discussion

The study highlighted and revealed that service availability and readiness of health facilities for common NCDs services were compromised in Ethiopia across various periods. The services available for diagnosis and management did not show improvement between 2014, 2016 and 2018 for diabetes (59%, 22% and 36%); for cardiovascular disease (73%, 41% and 49%); and chronic respiratory diseases (76%, 45% and 53%). Similarly, at the national level, the mean availability of tracer items between 2014, 2016 and 2018 was 37%, 53% and 48% for diabetes; was 36%, 41% and 42% for cardiovascular disease; was 26%, 27% and 27% for chronic respiratory diseases; and was 6%, 72% and 51% for cancer diseases. However, in 2014 survey year, the mean availability of tracer items was 7% each for mental illness and chronic renal diseases,

respectively. Similar findings have been documented in South Africa in 2014, which identified the need to improve availability of NCD-trained nurses, functional equipment, and medication [28]. Based on 2013 Ugandan study, none of the facilities included met the WHO package of essential non-communicable (PEN) standards for essential tools and medicines to implement effective NCD interventions [29].

In 2008 in Ethiopia that the overall prevalence of chronic NCDs was documented to be 8.9% and the specific prevalence were 0.5% for diabetes mellitus, 2.6% for hypertension, 3.0% for cardiovascular diseases, 1.5% for asthma and 2.7% for mental illness [30]. National cardiovascular disease prevalence was 15%, cancer and chronic obstructive pulmonary disease prevalence was 4% each, and diabetes mellitus prevalence was 2% [31].

Health facilities included in this study were not well equipped to provide services for diabetes, cardiovascular diseases, and chronic respiratory diseases. The trend was similar across the survey years. Diabetes had the lowest service availability, but the highest readiness score to provide service among the common NCDs. The availability of guidelines and at least one trained staff is minimal to deliver the management of diabetes, cardiovascular diseases, cervical cancer, and chronic respiratory disease services. According to the 2020 WHO report on the health care capacity for NCD management, evidence-based guidelines for the management of NCDs need to be developed and broadly implemented to ensure appropriate diagnosis, referral and treatment. Also reported was that, globally, only 48% of countries reported having evidence-based guidelines, protocols or standards for the management of all four NCDs through a primary care approach. Contrary to our study, the report revealed that guidelines for diabetes were most widely available (84%), followed by CVDs (77%), cancer (70%) and chronic respiratory diseases(CRDs) (64%). Correspondingly, countries were most likely to have guidelines for diabetes that included referral criteria and were utilized in at least 50% of health care facilities, with 80% of countries in upper-middle-income countries reporting having such guidelines [32]. The decline in the trends of service availability and readiness could be the fact that the skilled staff turnover is high form the rural to urban areas. Also, the primary health care system might not be ready in terms of infrastructure, staffing and medicine and commodities which is supposed to provide the service, and its inclusion increases the denominator when calculating readiness.

Our study is consistent with the study conducted in Uganda in 2012. Less than a half (46%) of the health facilities had hypertension guidelines with significant differences observed across health facility levels [33], which is one of the core aspects of the management of the chronic diseases. Similarly, a study done in Tanzania showed that in many health facilities, guidelines, diagnostic equipment, and first-line drug therapy for the primary care of NCDs were inadequate, and management, training, and reporting systems were weak [17]. Likewise, in the recent Tanzanian national survey, out of 725 health facilities involved in the study, only 28% of the assessed facilities were considered prepared for the outpatient primary care of hypertension. About 9% and 42% of the assessed facilities reported to have at least one trained staff and guidelines for hypertension, respectively [34]. In a Ghanaian study, the findings highlight the unavailability of essential medicines required for adequate management of NCDs reflected in the limited capacity at the various levels of care [35]. More importantly, there are challenges for healthcare delivery system: first the double burden of communicable diseases and non-communicable diseases such as diabetes and cardiovascular disease; the burden that chronic conditions impose on health systems and multiple chronic conditions; and lastly burden on the primary healthcare [36]. According to the WHO recommendations, training of the health workforce and strengthening the capacity of the health systems, particularly at primary care level and policy options to address the prevention and control of NCDs are all priorities [37].

From our findings, trained human resources were inadequate for addressing and managing NCDs in Ethiopia. Consistent result was documented in Tanzanian health facilities for outpatient primary care study, in which trained health workers for NCDs were inadequate, and management, training, and reporting systems were weak. Only one of the 24 health facilities surveyed reported any training in the past year for hypertension or diabetes [17]. According to the national survey in Tanzania, about 26% and 13% of health facilities had guidelines and trained staff for NCDs, respectively [38]. Coupled with this, in a Ghanaian study, inadequate staffing was a major impediment for the capacity of primary healthcare facilities to serve NCD-related health needs [35]. The capacity of the primary care system in Vietnam was inadequate to serve the NCD-related health needs of the population [39]. Similarly, a study documented in Ethiopia showed that encouraging actions and results on NCDs in terms of political commitment, however, there was a gap as shown by the limited availability of resources for NCDs prevention and treatment services at the primary health care level. Shortage of essential NCD drugs and diagnostic facilities and lack of treatment guidelines are major challenges [2].

This study showed that the majority of the services for chronic disease management occurred in a referral hospital, general hospital, and primary hospital, unlike the primary health care facilities. Although, people with, or at risk of developing, NCDs require long-term care that is proactive, patient- centered, community-based and sustainable. Such care can be delivered equitably only through robust health systems founded on strong primary health care towards the attainment of universal health coverage [32]. This is also the case in a 2013 survey from Uganda in that showed that limiting supply of anti-hypertensive medicines to higher-level health facilities was incongruent with the provision of high quality, chronic care for persons with hypertension in lower health facilities.

In the same way, lower-level health facilities, where the population is expected to receive primary health care, should be expected to stock essential medicines for treating non-communicable diseases [40]. A similar finding was also documented in Zambia where nearly all primary health facilities studied did not meet the minimum threshold to manage NCDs [41] according to the WHO recommendations [37].

The temporal trend identified for the common non-communicable disease service availability observed from 2014 to 2018 should be used to prioritize health care programs and scaling up interventions for NCDs in Ethiopia and other developing countries. Health system policy should give due attention to NCDs service availability. In addition, the government has to focus the attention towards prioritizing each specific disease. In addition, the government has to focus the attention towards prioritizing each specific disease (chronic respiratory diseases, cardiovascular diseases and diabetes) in accordance of disease burden while providing and expanding the NCDs service provision in government/public health facilities as well as focusing on rural area and primary health care services, where the majority of the community members are served.

Similarly, it will add value on the existing NCDs prevention strategies that would be strengthened through updating the priority disease that need further attention and implication.

Our study has some limitations. Firstly, the sampling frames used in the surveys was limited to health centers/health posts included in the country master health facility list and some other health facilities, especially those recently constructed, might not be included. Secondly, the cross-sectional nature of the study limits its value to explain the relative importance of some of the factors to be considered. At the same time, such kind of surveys can sometimes oversample hospitals and governmental and urban health facilities. Thirdly, another potential concern is the variations in the availability and management of NCDs across the regions of the country.

Lastly, p-values estimation should be regarded as exploratory rather than for testing hypothesis.

Despite the above limitations, our study has several strengths. First, this study was undertaken in a representative sample of health facilities, so it might be generalisable to all facilities in the country. Secondly, the survey tools used standardized WHO-recommended and validated data collection instrument, training. Furthermore, pre-testing of the tools was rigours in nature. The involvement and approval of policymakers and health programme managers supports the endorsement of the findings of the study for improving quality and management of NCDs carried out in the health facilities. In addition to this, the wealth of data allowed for looking at the temporal pattern across the years, which demonstrated the real situation of the health system's availability and readiness to provide NCD services in Ethiopia.

## Conclusions

The majority of the health facilities have low and gradual decrement in the availability of resources to provide NCD services in Ethiopia. However, there was stability and some increase in the readiness score across the some NCDs. However, more attention is needed on facility readiness to provide cervical cancer, mental illness, and chronic renal diseases services. There is a need to increase NCD service availability and readiness at the primary health care system, giving special attention to the primary hospitals and health centers, and private and rural health facilities where the majority of the population receive services. Basic diagnostic equipment and medication also need to be made available. Management of NCDs can also be improved by providing continuous training for primary health care workers. Opportunities for providing integrated care for NCDs should receive strong consideration, allowing for efficient use of the available resources at primary health care in low-income countries such as Ethiopia.

## Supporting information

**S1 Checklist. STROBE statement—checklist of items that should be included in reports of *cross-sectional studies*.**
(DOC)

## Acknowledgments

We acknowledged to the Ministry of Health and the Ethiopian Public Health Institute for providing the data. As well, we are highly indebted to appreciate the contribution of study participants as well as data collectors. The surveys were undertaken by the Ethiopian Public Health Institute (EPHI), which is the research arm of the Ethiopian Ministry of Health (FMoH). They were also done in collaboration with various organizations. We would like to thank, For ICF International provided technical assistance, the United States Agency for International Development (USAID), World Bank, Irish Aid, WHO and UNICEF for their support during the 2014 ESPA+. World Health Organization, World Bank and Global Fund for their support during the 2016 and 2018 surveys.

## Author Contributions

**Conceptualization:** Atkure Defar, Girum Taye Zeleke, Della Berhanu, Ephrem Tekle Lemango, Kassahun Alemu, Sibhatu Biadgilign.

**Data curation:** Atkure Defar, Girum Taye Zeleke.

**Formal analysis:** Atkure Defar, Girum Taye Zeleke, Della Berhanu, Ephrem Tekle Lemango, Sibhatu Biadgilign.

**Investigation:** Atkure Defar, Girum Taye Zeleke, Ephrem Tekle Lemango, Abebe Bekele, Kassahun Alemu, Sibhatu Biadgilign.

**Methodology:** Atkure Defar, Girum Taye Zeleke, Sibhatu Biadgilign.

**Project administration:** Atkure Defar.

**Resources:** Atkure Defar, Girum Taye Zeleke, Ephrem Tekle Lemango, Sibhatu Biadgilign.

**Software:** Girum Taye Zeleke.

**Supervision:** Atkure Defar, Girum Taye Zeleke, Della Berhanu, Ephrem Tekle Lemango, Abebe Bekele, Kassahun Alemu.

**Validation:** Atkure Defar, Girum Taye Zeleke, Della Berhanu, Ephrem Tekle Lemango, Abebe Bekele, Kassahun Alemu, Sibhatu Biadgilign.

**Visualization:** Atkure Defar, Girum Taye Zeleke, Della Berhanu, Sibhatu Biadgilign.

**Writing – original draft:** Atkure Defar, Sibhatu Biadgilign.

**Writing – review & editing:** Atkure Defar, Girum Taye Zeleke, Della Berhanu, Ephrem Tekle Lemango, Abebe Bekele, Kassahun Alemu, Sibhatu Biadgilign.

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
