## [Decision Letter · Decision Letter 0]

4 May 2022

PONE-D-22-03523Health System’s Availability and Readiness of Health Facilities for Chronic Non-Communicable Diseases: Evidence from the Ethiopian National SurveysPLOS ONE

Dear Dr. Biadgilign,

Thank you for submitting your manuscript to PLOS ONE. After careful consideration, we feel that it has merit but does not fully meet PLOS ONE’s publication criteria as it currently stands. Therefore, we invite you to submit a revised version of the manuscript that addresses the points raised during the review process. Specifically, the reviewers and I feel that the paper is important and useful but would benefit from revisions. The methods need to be clear in terms of how the tracer elements were decided for different NCDs and how the mean score was calculated . Reviewers have also commented about the statistical analysis. The differences among the type of facilities, or other background characteristics must be put in the context for readers to understand the variations. The discussion section also needs to be more concise clearly anlaysing the key results in a context of the country and then comparing with other countries.

We look forward to receiving your revised manuscript.

Kind regards,

Charu C Garg, Ph.D.

Academic Editor

PLOS ONE

Journal Requirements:

“The surveys were undertaken by the Ethiopian Public Health Institute (EPHI), which is the research arm of the Ethiopian Ministry of Health (FMoH). They were also done in collaboration with various organizations. For 2014 ESPA+, ICF International provided technical assistance and the survey was funded by the United States Agency for International Development (USAID), World Bank, Irish Aid, WHO and UNICEF. For 2016 and 2018 surveys, technical support was provided by the World Health Organization, World Bank and Global Fund. World Health Organization provided the financial support for the survey in 2016 and The World Bank provided financial support for the 2018 surveys.”

Additional Editor Comments (if provided):

Thanks for submitting this paper on an important area to this journal. The paper is interesting, but can benefit from some revisions as suggested by the 3 reviewers and also my own comments in the attached document.

The methods section needs to have more clarity on defining the tracer elements. Put it in the context of WB SDI and explain the reasons for variables that were chosen as tracer values. In the tables the n values must be provided when the % are being presented for each variable.

The discussion section looks a little haphazard. It might benefit by putting the key results in a context in the first para.. The results from other countries may be put in a next para. Followed by limitation strengths and confusion.

The abstract also needs to me made more crisp and requires a sentence in methods about how readiness and service availability are defined. Please see comments from other reviewer also in the attached document to complete your revision.

Reviewers' comments:

Reviewer's Responses to Questions

**Comments to the Author**

1. Is the manuscript technically sound, and do the data support the conclusions?

Reviewer #1: Yes

Reviewer #2: Yes

Reviewer #3: Yes

2. Has the statistical analysis been performed appropriately and rigorously? 

Reviewer #1: Yes

Reviewer #2: Yes

Reviewer #3: No

3. Have the authors made all data underlying the findings in their manuscript fully available?

Reviewer #1: No

Reviewer #2: Yes

Reviewer #3: Yes

4. Is the manuscript presented in an intelligible fashion and written in standard English?

Reviewer #1: Yes

Reviewer #2: Yes

Reviewer #3: No

5. Review Comments to the Author

Reviewer #1: The paper covers changes in health facility and readiness in managing NCDs in Ethiopia using national SARA surveys and interestingly reports a decline in many parameters. While potentially relevant for readers, it needs to address some critical issues better.

Abstract: The results especially the for availability of tracer items need to be given for each disease by year for easy understanding.

Main paper:

Introduction: 40 million people were not reported to have died from NCDs but were estimated to have died. Premature deaths due to NcDs are those between 30-69 years and not below 70 years.

As the paper focuses on comparison between the three surveys, it is critical that reader knows the three survey methodology better. It will be good to tabulate details of the three surveys ( sample size, sampling, tools used, other relevant measurement issues etc,).

The number and proportion of health facilities changed in the surveys in terms of primary/secondary/tertiary. Could these have affected the results and the decline is a reflection of choice of facilities than the decline. The authors have to ensure that measurement issues are not causing a bias in terms of decline of service availability and readiness. The results should also be presented by the level of facilities. Have they used a weighted prevalence to adjust for these differences?

Readiness was only measured in subset of facilities that offered services. I think the denominator has to be consistent as all facilities. It is not clear how the readiness score can be that half of the facilities that offered the service had each of the input. OR was it that 1tleast 50% of the inputs were available in a facility which had a readiness score of 50?

Table 1 - Region can be deleted. If needed a map may be provided with location of studies facilities to ensure that these are geographically well spread. As mental health and CKD were only seem to have been assessed in 2014, these can be deleted.

Discussion has to really focus on whether the decline is real or because of selection and measurement bias. If real, then a better explanation of the why this occurred based on the authors knowledge of context. There is no point comparing these findings with other countries as they are not very relevant.

Reviewer #2: This is an important analysis in the given context and the use of standardised survey data makes it comparable across countries.

1. The discussion could be deepened to reflect also on why there is a downward trend in service availability and correlated with other literature from the same setting.

2. Limitations can include the lack of any explanations for the trends observed- which would have been possible with a qualitative component to explore why some of these findings are seen.

3. The service availability and readiness seem to show opposite trends - please comment and discuss

4. Discussion on type of facility and service availability- higher facilities are expected to have more availability but primary health care strengthening is of importance in the management of chronic conditions, this requires some discussion and comment

Reviewer #3: The authors report on 3 rounds of a national survey of public sector health facilities to understand the overall availability and readiness to provide care for the growing prevalence of a range of NCDs (HTN, DM, cancer and mental health). The use of the existing data is a clear strength for the manuscript, reflecting large gaps in availability and in readiness with limited improvement over time. The paper is particularly important given the growing burden of NCDs in Nigeria, the region and globally. However the paper needs revisions to be able to better communicate the results, expand on what is presented and strengthening the discussion.

Introduction

The introduction is solid It would be good to understand how in the presence of rising prevalence and low system readiness, the NCD-related mortality also dropped. Also would be good to discuss in the discussion. It would also be important to frame the paper and readiness results as necessary but not sufficient for quality coverage for NCDs.

Methods

Was there a need to adjust for sampling since some analyses were done looking at facility type and if a national estimate is needed?

Why was 50% determined to be the threshold?

Is there a cross walk between the areas covered and methods in the different survey types?

Data collection-states “adapted version” of ESPA+ and SDi-which version and is there a reference?

Which results were used-interview or observation?

Under ethical-what is meant by “sometimes health workers responsible for client services”

Results:

The authors are very detailed in providing rates of availability and readiness, but largely repeat what is in the tables. However there are areas of details which would significantly strengthen the impact of this paper. For example, exploring where were areas of gaps across sites and going into more details of the readiness areas rather than just the mean. In addition, the table 1is hard to understand with numbers versus % and needs definitions of the facility types. The authors also note subnational differences, and so more details such as urban/rural and in the discussion why state-level differences may have been seen. This is particularly important as discussed in discussion. Statistics should be included in the results, not just the table (and should be included in the figure)-for example page 10 where it states “statistically significant”

Some analysis of overall access (availability plus readiness would also strengthen the results

Were there any tests for trends of availability and readiness? Versus T tests?

Page 8 bottom-“the sentence starting with Only cancer diseases and mental illness and chronic renal diseases services were diagnosed and managed….” Is confusing and need clarification

Page 9: states bar graph shows trend for 3 NCDs, but then lists 4

Discussion

The discussion at the start is repeating results-It would be more effective to summarize broadly and then dive into the similarities and differences with previous studies. In the discussion of rising burden . In limitations-would also include that private sector was not included (by my understanding). While the policy implications is a good approach-the paragraph is a little hard to follow and could be simplified for a stronger message-perhaps-rising rates of NCDs, no real change in readiness-policy, resources need to focus.

Minor-please review for sentence and grammar-such as the first sentence (readiness and readiness”. The manuscript particularly the discussion would be improved by a careful review and edit

6. PLOS authors have the option to publish the peer review history of their article (what does this mean?). If published, this will include your full peer review and any attached files.

Reviewer #1: **Yes: **Anand Krishnan

Reviewer #2: No

Reviewer #3: No

---

## [Author Response · Author response to Decision Letter 0]

28 Jul 2022

Dear Oriel Jerome Delas Alas Vida

PLOS ONE

Greetings of the days! Please kindly find below response for the request made during the review process. I hope this will enable to further process the manuscript. 

a) Please indicate the ethical or legal reasons that have led the data to be restricted (e.g., data contain potentially identifying or sensitive patient information).

• There is no ethical or legal reason, but it's all about ownership. Originally, the data owner is the "National Data Management Center (NDMC)" at Ethiopian Public Health Institute (EPHI). National health and health related data have their own data governance system and shall be managing under the national data sharing policy. We have included the links to access the data. Please see below the links posted. 

b) Include the name of the ethics committee, Institutional Review Board, or other organization that is imposing the restriction.

• The Ethiopian Public Health Institute (EPHI). The institute has its own data repository system at the NDMC, anyone interested to use the data sets could communicate to the center using the following link; https://rtds.ephi.gov.et/public/showdetail/94#step-1

c) Provide a non-author, institutional point of contact (and contact information) that is able to field data access queries. PLOS ONE's data policy requires this in the interest of maintaining long-term data accessibility.

• The data can be accessed using following URL- ESPA+ 2014 (https://rtds.ephi.gov.et/public/showdetail/76), SARA 2016 (https://rtds.ephi.gov.et/public/showdetail/93), SARA 2018 (https://rtds.ephi.gov.et/public/showdetail/94)

d) If relevant, add any data set names, variables, accession codes, URLs, DOIs, etc. that an independent researcher would need in order to request your minimal data set.

• The data are available upon request from the National Data Management Center (NDMC), EPHI website but are not available in public repository because the original data set is owned by NDMC) not by the authors. Anyone may contact or visit NDMC at https://rtds.ephi.gov.et/public/showdetail/94#step-1 to access the data.

With regards,

---

## [Decision Letter · Decision Letter 1]

13 Sep 2022

PONE-D-22-03523R1Health System’s Availability and Readiness of Health Facilities for Chronic Non-Communicable Diseases: Evidence from the Ethiopian National SurveysPLOS ONE

Dear Dr. Biadgilign,

Thank you for submitting your manuscript to PLOS ONE. After careful consideration, we feel that it has merit but does not fully meet PLOS ONE’s publication criteria as it currently stands. Therefore, we invite you to submit a revised version of the manuscript that addresses the points raised during the review process. While 2 of the reviewers feel that the revised paper can be accepted, one of them still has reservations. My major reservation is on abstract and discussion section, which still needs to be improved. Also I think that the clean version does not show all changes made in track change mode. Please try to be consistent across the versions. My specific comments are attached below. 

We look forward to receiving your revised manuscript.

Kind regards,

Charu C Garg, Ph.D.

Academic Editor

PLOS ONE

Journal Requirements:

Additional Editor Comments (if provided):

While the paper has been greatly improved, I find that all the changes made in the track changes version, did not show up in the clean version. Please upload correct versions. I think that the reviewer that has made comments specifically about sampling etc. did not see the track change mode but the clean version, where those changes are missed. 

1. Agree with the reviewer that results in abstract are very descriptive with just the numbers. Can we highlight the major findings under the results. The % may only be presented for 2018 and show where there are improvements from previous years and why? Also focus on the types of facility and regional differences to bring about the difference in performance.

2. Number of facilities is not important in the methods in abstract. Can you briefly say some tracer elements, especially which are significant in your study and also discuss these under results in the abstract.

3. Needs editing – eg. Ethiopia repeated twice in abstract under conclusions; Para 1 in introduction;

4. It is mentioned that the analysis was done across types of facilities, residence, region – the differentials for both availability and readiness across different facilities should be more clearly highlighted. There is only one small discussion about training in the discussion across the facilities.

5. I feel discussion needs to be rearranged. Not sure about the significance of giving prevalence in beginning of the 2nd para of discussion. That needs to be placed in a context. As also mentioned by one of the reviewers it would be useful to critically discuss the results and state why availability or readiness is low and when it improved, why and if not, why not. You could also consider organizing the discussion around availability and readiness or by diseases. Under each of these clearly organize which tracer elements are strengthened. And at what level of facility. Too much of repetition after every sentence about Uganda, Ghana and Zambia, can you first discuss your major findings and then say what these other studies say about your results. Might make it easier to follow.

Reviewers' comments:

Reviewer's Responses to Questions

**Comments to the Author**

1. If the authors have adequately addressed your comments raised in a previous round of review and you feel that this manuscript is now acceptable for publication, you may indicate that here to bypass the “Comments to the Author” section, enter your conflict of interest statement in the “Confidential to Editor” section, and submit your "Accept" recommendation.

Reviewer #1: All comments have been addressed

Reviewer #2: All comments have been addressed

Reviewer #3: (No Response)

2. Is the manuscript technically sound, and do the data support the conclusions?

Reviewer #1: Yes

Reviewer #2: Yes

Reviewer #3: Partly

3. Has the statistical analysis been performed appropriately and rigorously? 

Reviewer #1: Yes

Reviewer #2: Yes

Reviewer #3: I Don't Know

4. Have the authors made all data underlying the findings in their manuscript fully available?

Reviewer #1: Yes

Reviewer #2: Yes

Reviewer #3: Yes

5. Is the manuscript presented in an intelligible fashion and written in standard English?

Reviewer #1: Yes

Reviewer #2: Yes

Reviewer #3: No

6. Review Comments to the Author

Reviewer #1: (No Response)

Reviewer #2: (No Response)

Reviewer #3: Review PLOS

The authors have leveraged existing facility assessment data to highlight the lack of progress (or even decline) in facility readiness to treat the rising burden of NCDs in Ethiopia. The analysis to increase the use of these data is to be applauded, but the paper as written needs strengthening to better convey and discuss the findings. I have included some broader areas as well as some details where edits are needed which would make the paper stronger (as well as a review for grammar-long sentences, a page and ½ paragraph in discussion etc.)

1. Abstract

a. In the abstract, only listing the numbers make it hard to read in the medication readiness. Reorganizing to show persistently low by area would be better, although some may have improved albeit remaining low For example levels of medications remained low between 2014 and 2018 for Diabetes (____ to ___%), cardiovascular disease (____ to ____) etc.

b. Were there any statistics done to see of these were insignificant changes or if some actually increase

2. Introduction

a. The statement about NCD deaths in Ethiopia needs a reference

b. I am also confused by the sentence” The Disability Adjusted Life Years (DALYs) increased from below 20% in 1990 to 69% in 2015 and the incidence and prevalence of NCDs are become rising”

c. The first paragraph gets a bit repetitive and hard to follow esp. as all about increase but the final sentence states “The death rates of NCDs causes declined by 40% from 1990 to 2015 and also from the 680.9 per 100,000 crude all-cause mortality rate, 286.9 (95% UI: 188.1–423.0) per 100,000 people” which is confusing

d. The sentence which is long on other studies “These studies, however, are limited as they focus on a particular or selected set of NCDs 18-20.Other studies in the country also showed that from a total 21 public health facilities surveyed, only 38.1% had all the categories of health professionals as compared to the national standards and 85.2%, of the facilities fulfilled the criteria for basic equipment and 47.7% of the facilities did not fulfill the criteria for infection prevention supplies21 and this is linked with the availability of essential equipment and drugs, health workers' communication, health care provided, and attitude of health workers had positive association with client satisfaction with various health programs in the health system” is hard to understand-is this about NCDs

e. The last sentence highlights other gaps than those covered in the study-so would be good to explain which of the gaps this study described

3. Methods

a. How did the surveys in 2014 versus 2016 and 18 differ (if at all). Were the sampling different (a little hard to tell)

b. Were private facilities included (if not need to be clear this is only about public facilities)

c. What language was used ? If translated how was that done?

d. What does the references for “The availability of the services and readiness of health facilities for diabetes, cardiovascular, and chronic respiratory diseases were investigated in the ESPA+ 2014, and the two recent (2016 and 2018) SARA surveys”- was this referencing for the definitions. I would also think that referral does not constitute availability. Would consider a sub analysis removing that option. Also this is not consistent with what is stated in the following section “We defined facility ‘'availability" as the percentage of facilities in the sample that said they offered the service in question, which considered whether the diagnosis and/or management services were available. The results section is also confusing as seems like a broader set of requirements for availability

e. The description of calculation of readiness was a little hard to follow-is readiness a dichotomous outcome and so as stated 50% means 50% are ready but the following statement notes” A readiness score of 50 signifies that, on average, half of the facilities that offered the service had each of the requisite inputs for delivering that service. The calculated indices were compared to an agreed cutoff point of 50%. A facility index that was below 50% was considered as `not ready' to manage NCDs.” Can this be clarified

f. How were missing data handled if any)

4. Results

a. Table 1 needs some work-it says % but some are >100%

i. Some of the facility types are not described and if health posts are not included-which would these be?

b. Were there any differences in facilities studied between the years? As the number dropped

c. How were the p values determined (table 3) and seems like these should be referenced in statements are made about decrease (or not) and if and how adjusting for facility type (ex for differences based on rural/urban). For Table.3 I assume this is absolute change (but needs description in methods and table). Page 10 also state the p value not just say significant

d. Given the range in the different components of the readiness, a discussion of where the areas that remain low versus those which are stronger may help

e. Some information should be first presented /reported in results beyond the table versus discussion -such as difference in availability in higher level facilities-would consider decreasing words in methods and expand results. Same for sector differences if any existed

f. Table 2:

i. Is the mean the average of the average of total facility score?

5. Discussion

a. First sentence need editing and in the first paragraph would not just report the details, but have a broader summary-where were areas of particular weakness within and across NCDs for example. Similar comments are throughout the discussion. Some of this is found in the overly long paragraph (page 11-12) but should be a separate paragraph better explained

b. It is curious that readiness for DM is highest but availability is lowest. In the discussion this should be discussed and what this means in terms of accessibility (which facilities had availability and hence were measured for readiness-more urban? Higher level?

c. Would consider a measure of access-which would what % of all facilities had the care available and adequate readiness-I suspect the % would be even lower

d. The PEN standards are important, but in bringing them in, a brief description of if and how they differ from the standards used in the study are needed

e. For limitations-what does “seemed” mean for oversampled?

f. In the last paragraph before conclusions-the first sentence is a it repetitive. Also is the problem policy or funding of the policy or funding but supply and training? I did not see that discussed. Th conclusion is also. A bit repetitive and some recommendations are perhaps not as strongly supported (ex. Training-is this a response to trained HCWs-or is the gap in actual number of HCWs-the deficits in materials also seems to be the most important bottleneck?

7. PLOS authors have the option to publish the peer review history of their article (what does this mean?). If published, this will include your full peer review and any attached files.

Reviewer #1: **Yes: **Anand Krishnan

Reviewer #2: No

Reviewer #3: No

---

## [Author Response · Author response to Decision Letter 1]

2 Mar 2023

Rebuttal Letter 

Dear Charu C Garg (Ph.D)

Academic Editor

PLOS ONE Journal 

Thanks for the comments, concerns, and revision points you and the three reviewers have provided to this manuscript titled “Health System’s Availability and Readiness of Health Facilities for Chronic Non-Communicable Diseases: Evidence from the Ethiopian National Surveys” number PONE-D-22-03523R1. We have now addressed most of the comments and amended, you could find in the file “revised manuscript track changes” and highlighted in red. And we have revised the manuscript according to the PLOS ONE style and guideline and captions are corrected. 

Additional Editor Comments (if provided):

While the paper has been greatly improved, I find that all the changes made in the track changes version, did not show up in the clean version. Please upload correct versions. I think that the reviewer that has made comments specifically about sampling etc. did not see the track change mode but the clean version, where those changes are missed. 

1. Agree with the reviewer that results in abstract are very descriptive with just the numbers. Can we highlight the major findings under the results? The % may only be presented for 2018 and show where there are improvements from previous years and why? Also focus on the types of facility and regional differences to bring about the difference in performance.

Response: It is a good comment suggested regarding the numbers depicted in the result section of the abstract. We present the finding in the form with types of facilities, residence, and region as described in Table 1. However, we are not compared and analyzed to find out the differentials for both availability and readiness across different facilities, region and residence, and this is not the scope of the paper we are addressed.

2. Number of facilities is not important in the methods in abstract. Can you briefly say some tracer elements, especially which are significant in your study and also discuss these under results in the abstract?

Response: It is a good comment and revised in the current file. 

3. Needs editing – e.g. Ethiopia repeated twice in abstract under conclusions; Para 1 in introduction;

Response: It is a good comment and revised in the current file. 

4. It is mentioned that the analysis was done across types of facilities, residence, region – the differentials for both availability and readiness across different facilities should be more clearly highlighted. There is only one small discussion about training in the discussion across the facilities.

Response: It is a good comment and revised in the current file. We present the finding in the form with types of facilities, residence, and region as described in Table 1. However, we are not compared and analyzed to find out the differentials for both availability and readiness across different facilities, region and residence, and this is not the scope of the paper we are addressed. 

5. I feel discussion needs to be rearranged. Not sure about the significance of giving prevalence in beginning of the 2nd para of discussion. That needs to be placed in a context. As also mentioned by one of the reviewers it would be useful to critically discuss the results and state why availability or readiness is low and when it improved, why and if not, why not. You could also consider organizing the discussion around availability and readiness or by diseases. Under each of these clearly organize which tracer elements are strengthened. And at what level of facility. Too much of repetition after every sentence about Uganda, Ghana and Zambia, can you first discuss your major findings and then say what these other studies say about your results. Might make it easier to follow.

Response: We have organized the paper as per the reviewers and editor feedback to improve the quality of the paper. We removed the prevalence aspect in the discussion section. 

Reviewers' comments:

Reviewer's Responses to Questions

Comments to the Author

1. If the authors have adequately addressed your comments raised in a previous round of review and you feel that this manuscript is now acceptable for publication, you may indicate that here to bypass the “Comments to the Author” section, enter your conflict of interest statement in the “Confidential to Editor” section, and submit your "Accept" recommendation.

Reviewer #1: All comments have been addressed

Reviewer #2: All comments have been addressed

Reviewer #3: (No Response)

Response: It is a good comment and appreciates the reviewers thought to improve the manuscript. 

2. Is the manuscript technically sounds, and do the data support the conclusions?

Reviewer #1: Yes

Reviewer #2: Yes

Reviewer #3: Partly

Response: Thank you for your comment. 

3. Has the statistical analysis been performed appropriately and rigorously?

Reviewer #1: Yes

Reviewer #2: Yes

Reviewer #3: I Don't Know

Response: Thank you for your comment and do acknowledge the feedback. 

4. Have the authors made all data underlying the findings in their manuscript fully available?

Reviewer #1: Yes

Reviewer #2: Yes

Reviewer #3: Yes

Response: Thank you for your feedback. 

5. Is the manuscript presented in an intelligible fashion and written in standard English?

Reviewer #1: Yes

Reviewer #2: Yes

Reviewer #3: No

Response: Thank you for your comment and do acknowledge the feedback. 

6. Review Comments to the Author

Reviewer #1: (No Response)

Reviewer #2: (No Response)

Reviewer #3: Review PLOS

The authors have leveraged existing facility assessment data to highlight the lack of progress (or even decline) in facility readiness to treat the rising burden of NCDs in Ethiopia. The analysis to increase the use of these data is to be applauded, but the paper as written needs strengthening to better convey and discuss the findings. I have included some broader areas as well as some details where edits are needed which would make the paper stronger (as well as a review for grammar-long sentences, a page and ½ paragraph in discussion etc.)

Response: Thank you for your comment and do acknowledge the feedback. 

1. Abstract

a. In the abstract, only listing the numbers makes it hard to read in the medication readiness. Reorganizing to show persistently low by area would be better, although some may have improved albeit remaining low For example levels of medications remained low between 2014 and 2018 for Diabetes (____ to ___%), cardiovascular disease (____ to ____) etc.

Response: Thank you for your comment and do acknowledged the feedback. 

b. Were there any statistics done to see of these were insignificant changes or if some actually increase

Response: We use Fisher's exact test was used to assess the difference in the mean availability of tracer items as we see in Table 3. 

2. Introduction

a. The statement about NCD deaths in Ethiopia needs a reference

Response: Thank you for your comment and amended accordingly. 

b. I am also confused by the sentence” The Disability Adjusted Life Years (DALYs) increased from below 20% in 1990 to 69% in 2015 and the incidence and prevalence of NCDs are become rising”

Response: Thank you for your comment. Both the prevalence and incidence of NCDs getting higher in the country. The evidence generated from Global Burden of Disease indicates this circumstance in more detailed manner (i.e. https://pubmed.ncbi.nlm.nih.gov/28732542/, https://www.ajol.info/index.php/ejhd/article/view/178817)

c. The first paragraph gets a bit repetitive and hard to follow esp. as all about increase but the final sentence states “The death rates of NCDs causes declined by 40% from 1990 to 2015 and also from the 680.9 per 100,000 crude all-cause mortality rate, 286.9 (95% UI: 188.1–423.0) per 100,000 people” which is confusing.

Response: Thank you for your comment and the statement is now clear in the amended manuscript accordingly. 

d. The sentence which is long on other studies “These studies, however, are limited as they focus on a particular or selected set of NCDs 18-20.Other studies in the country also showed that from a total 21 public health facilities surveyed, only 38.1% had all the categories of health professionals as compared to the national standards and 85.2%, of the facilities fulfilled the criteria for basic equipment and 47.7% of the facilities did not fulfill the criteria for infection prevention supplies21 and this is linked with the availability of essential equipment and drugs, health workers' communication, health care provided, and attitude of health workers had positive association with client satisfaction with various health programs in the health system” is hard to understand-is this about NCDs

Response: It is a good comment and we removed the statement from the body of the manuscript. 

e. The last sentence highlights other gaps than those covered in the study-so would be good to explain which of the gaps this study described

Response: The gap identified in other studies is basically focused on neither specific disease entity nor items. So, our study conducted in a comprehensive manner addressing a wide range from the availability of guideline and essential drugs, training, logistic and equipment. 

3. Methods

a. How did the surveys in 2014 versus 2016 and 18 differ (if at all). Were the sampling different (a little hard to tell)

Response: It is a good comment and there is no difference among the surveys methodology that differ for the interpretation of the finding. However, we described the approach and methods used in more detailed manner in the method section of the manuscript. To be more elaborative, in 2014 sampling, 

all hospitals were included and a representative sample of Health Centers, Health posts and Clinics were included. While, in 2016 sampling, all hospitals were included and a representative sample of Health Centers, Health posts and Clinics were included and in 2018 sampling, all hospitals were included and a representative sample of Health Centers, Health posts and Clinics were included. These imply, no sampling difference in the surveys. 

b. Were private facilities included (if not need to be clear this is only about public facilities?)

Response: We have included private facilities and described in Sampling of health facilities part in method section. In addition, in both SPA 2014, SARA 2016 and SARA 2018 the managing authority for the Clinics are private. 

c. What language was used ? If translated how was that done?

Response: In SPA 2014, the standard English questionnaire was translated to the three local languages: Amharic, Afaan Oromo and Tigrigna while in SARA 2016, the standard English questionnaire was translated to Amharic and lastly in SARA 2018, the standard English questionnaire was translated to Amharic. 

d. What does the references for “The availability of the services and readiness of health facilities for diabetes, cardiovascular, and chronic respiratory diseases were investigated in the ESPA+ 2014, and the two recent (2016 and 2018) SARA surveys”- was this referencing for the definitions. I would also think that referral does not constitute availability. Would consider a sub analysis removing that option. Also this is not consistent with what is stated in the following section “We defined facility ‘'availability" as the percentage of facilities in the sample that said they offered the service in question, which considered whether the diagnosis and/or management services were available. The results section is also confusing as seems like a broader set of requirements for availability.

Response: The scope of the study and the data used is coming from a national survey conducted using WHO standard approach used widely in the globe for assessing availability and readiness of service provision. All in all, the method and definitions are also following the same standard procedures. 

it refers as a definition for “availability of services for non-communicable diseases”. And in the sentence “The service was considered to be available if a facility manages, diagnoses, treats or refers a patient coming to the facility for common NCDs.” The word refers inserted by mistake. It needs to be replaced by “prescribe”

e. The description of calculation of readiness was a little hard to follow-is readiness a dichotomous outcome and so as stated 50% means 50% are ready but the following statement notes” A readiness score of 50 signifies that, on average, half of the facilities that offered the service had each of the requisite inputs for delivering that service. The calculated indices were compared to an agreed cutoff point of 50%. A facility index that was below 50% was considered as `not ready' to manage NCDs.” Can this be clarified

Response: Readiness score is not a dichotomous outcome. For instance, in SARA 2016, related to “Cervical cancer”, 53% of facilities had guideline, 61% of facilities had at least 1 trained staff, 96% of facilities had speculum, 77% of facilities had acetic acid (These are required inputs/items to provide cervical cancer services). It implies the mean availability of these 4 tracer items readiness score will be ([53%+61%+96%+77%]/4)=72%. It reveals this facility was ready to provide cervical cancer service (>50%). 

f. How were missing data handled if any)

Response: If there is missing data, it was excluded from analysis 

4. Results

a. Table 1 needs some work-it says % but some are >100%

i. Some of the facility types are not described and if health posts are not included-which would these be?

Response: the health posts are removed from this study as they did not provide the services and already described in the method section under statistical analysis. 

b. Were there any differences in facilities studied between the years? As the number dropped

Response: of course, the sample we use from the list of health facilities varies across the year. 

c. How were the p values determined (table 3) and seems like these should be referenced in statements are made about decrease (or not) and if and how adjusting for facility type (ex for differences based on rural/urban). For Table.3 I assume this is absolute change (but needs description in methods and table). Page 10 also state the p value not just say significant

Response: it is a good point and we have amended that in the method section.

d. Given the range in the different components of the readiness, a discussion of where the areas that remain low versus those which are stronger may help

Response: it is a good comment and updated accordingly. 

e. Some information should be first presented /reported in results beyond the table versus discussion -such as difference in availability in higher level facilities-would consider decreasing words in methods and expand results. Same for sector differences if any existed

Response: there is no analysis carried out to compare health facilities in which it is not the aim of the study. However, the authors believe from experience that the hospitals provide more service for NCD in terms of availability and readiness to provide the intended services than health centers.

f. Table 2:

i. Is the mean the average of the average of total facility score?

Response: it is a good point and it is not like that. The mean availability of tracer items for each disease that are described in the table and their tracer element is the one to be measured. 

5. Discussion

a. First sentence needs editing and in the first paragraph would not just report the details, but have a broader summary-where were areas of particular weakness within and across NCDs for example. Similar comments are throughout the discussion. Some of this is found in the overly long paragraph (page 11-12) but should be a separate paragraph better explained

Response: the first paragraph outlined the summery of the main finding. But, we tried to shorten the paragraphs accordingly in the revised version of the manuscript. 

b. It is curious that readiness for DM is highest but availability is lowest. In the discussion this should be discussed and what this means in terms of accessibility (which facilities had availability and hence were measured for readiness-more urban? Higher level?

Response: approximately the service availability and readiness correspond for the service for diabetes mellites. However, the service availability and readiness for DM is higher in urban and facility type where most of the service is higher for availability and readiness in hospital than PHC setting in the country. 

c. Would consider a measure of access-which would what % of all facilities, had the care available and adequate readiness-I suspect the % would be even lower

Response: It is out of the scope of the methodology for assessing the health facilities in terms of accessibility. 

d. The PEN standards are important, but in bringing them in, a brief description of if and how they differ from the standards used in the study are needed

Response: It is a good point. However, the PEN standard also follow the same approach that the SARA and SPA measurement used as a modality of checking the and strengthen the health services delivery and the management of the PHC system for diagnosis and treatment of selected NCDs. 

e. For limitations-what does “seemed” mean for oversampled?

Response: We have removed the wording, it is typo error. 

f. In the last paragraph before conclusions-the first sentence is a it repetitive. Also, is the problem policy or funding of the policy or funding but supply and training? I did not see that discussed. Th conclusion is also. A bit repetitive and some recommendations are perhaps not as strongly supported (ex. Training-is this a response to trained HCWs-or is the gap in actual number of HCWs-the deficits in materials also seems to be the most important bottleneck?

Response: It is a good point. In the conclusion we are summarizing the few bullet points of what is actual conclude from this study. The training is to support the HCW technical capacity in managing the patients. As suggested from the reviewer, logistic and materials availability for treating cases is also important and addressed in the revised version of the manuscript. 

7. PLOS authors have the option to publish the peer review history of their article (what does this mean?). If published, this will include your full peer review and any attached files. Do you want your identity to be public for this peer review? For information about this choice, including consent withdrawal, please see our Privacy Policy.

Reviewer #1: Yes: Anand Krishnan

Reviewer #2: No

Reviewer #3: No

Best regards,

---

## [Decision Letter · Decision Letter 2]

11 Apr 2023

PONE-D-22-03523R2Health System’s Availability and Readiness of Health Facilities for Chronic Non-Communicable Diseases: Evidence from the Ethiopian National SurveysPLOS ONE

Dear Dr. Biadgilign,

Thank you for submitting your manuscript to PLOS ONE. After careful consideration, we feel that it has merit but does not fully meet PLOS ONE’s publication criteria as it currently stands. Therefore, we invite you to submit a revised version of the manuscript that addresses the points raised during the review process.

The comments have been well addressed in the revisions except a few pointed out by the reviewers, specifically related to table formats, grammatical errors and a few explanations. If some of the things cannot be explained, they may be stated in the limitations. 

We look forward to receiving your revised manuscript.

Kind regards,

Charu C Garg, Ph.D.

Academic Editor

PLOS ONE

Journal Requirements:

Additional Editor Comments:

he paper has been substantially improved. There are minor comments especially related to table formats, grammatical errors and a few explanations. Once those are addressed, the paper can be accepted,

Reviewers' comments:

Reviewer's Responses to Questions

**Comments to the Author**

1. If the authors have adequately addressed your comments raised in a previous round of review and you feel that this manuscript is now acceptable for publication, you may indicate that here to bypass the “Comments to the Author” section, enter your conflict of interest statement in the “Confidential to Editor” section, and submit your "Accept" recommendation.

Reviewer #3: (No Response)

Reviewer #4: (No Response)

2. Is the manuscript technically sound, and do the data support the conclusions?

Reviewer #3: Yes

Reviewer #4: (No Response)

3. Has the statistical analysis been performed appropriately and rigorously? 

Reviewer #3: Yes

Reviewer #4: I Don't Know

4. Have the authors made all data underlying the findings in their manuscript fully available?

Reviewer #3: Yes

Reviewer #4: (No Response)

5. Is the manuscript presented in an intelligible fashion and written in standard English?

Reviewer #3: No

Reviewer #4: Yes

6. Review Comments to the Author

Reviewer #3: The authors have worked to respond to the comments. The paper still requires a careful read for typos and grammar. For example, in the abstract, “Proposition of facilities that

offered the service for diabetes, cardiovascular disease, chronic respiratory disease, cancer

diseases, mental illness, and chronic renal diseases was calculated to measure health service

availability” should be corrected to proportion. In the introduction “and the incidence and prevalence of NCDs are become rising”

Abstract: I was not sure what “However, in 2014 , the mean availability of

tracer items was 7% each for mental illness and chronic renal diseases, each respectively” signified-what were the results in more recent years? Same comment for discussion

Methods: I remain confused about “A readiness score

of 50 signifies that, on average, half of the facilities that offered the service had each of the

requisite inputs for delivering that service. The calculated indices were compared to an agreed

cutoff point of 50%. A facility index that was below 50% was considered as `not ready' to

manage NCDs.”. This seems to imply use overall and then later in the sentence about an individual facility. There for is a score of 40% that 40% of facilities were above 50%??

Results- Statistically significant is not adequate-you need to put in the results in the text

The number of % of facility by type seems to have changed a lot across the surveys-with many more lower level facilities early on. How did the authors adjust and if not, need to add to limitations that some change could be due to different make-up of the sample

In the discussion, please include the dates for the surveys being used in comparisons (South Africa and Uganda). Same for the data from Ethiopia

Table 1: Would be easier to interpret if the columns included both % and n rather than having the reader calculate the number of facilities

Table 3: Would limit p values to 2 decimal points if not significant. I also assume difference is absolute, nor relative but that should be made clear

Reviewer #4: Substantial revisions have been made from the feedback of previous reviewers. I wish to however draw your retention to table 1. In the last row of totals, it is not clear what it represents. Attempt to add urban and rural located facilities gives conflicting totals from what is imputed in most of the cells. It may also be important to give a bit more details about the table as a footnote, including spelling out abbreviations/acronyms so that the table can be understandable if standing alone (this should also apply to other tables/figures). it may also be important to include the denominator number for each characteristic being measured for a better understanding of the proportions.

7. PLOS authors have the option to publish the peer review history of their article (what does this mean?). If published, this will include your full peer review and any attached files.

Reviewer #3: No

Reviewer #4: No

---

## [Author Response · Author response to Decision Letter 2]

16 Oct 2023

Rebuttal Letter 4

Dear Charu C Garg (Ph.D)

Academic Editor

PLOS ONE Journal 

Thanks for the comments, concerns, and revision points you and the three reviewers have provided to this manuscript titled “Health System’s Availability and Readiness of Health Facilities for Chronic Non-Communicable Diseases: Evidence from the Ethiopian National Surveys” number PONE-D-22-03523R1. We have now addressed most of the comments and amended, you could find in the file “revised manuscript track changes” and highlighted in red. And we have revised the manuscript according to the PLOS ONE style and guideline and captions are corrected. 

Journal Requirements:

Additional Editor Comments:

he paper has been substantially improved. There are minor comments especially related to table formats, grammatical errors and a few explanations. Once those are addressed, the paper can be accepted,

Reviewers' comments:

Reviewer's Responses to Questions

Comments to the Author

Question 1. If the authors have adequately addressed your comments raised in a previous round of review and you feel that this manuscript is now acceptable for publication, you may indicate that here to bypass the “Comments to the Author” section, enter your conflict of interest statement in the “Confidential to Editor” section, and submit your "Accept" recommendation.

Reviewer #3: (No Response)

Reviewer #4: (No Response)

Response: It is a good reflection and thank you for the feedback. 

Question 2. Is the manuscript technically sound, and do the data support the conclusions?

Reviewer #3: Yes

Reviewer #4: (No Response)

Response: Thank you for the feedback and for the positive impression. 

Question 3. Has the statistical analysis been performed appropriately and rigorously?

Reviewer #3: Yes

Reviewer #4: I Don't Know

Response: Thank you for the feedback and for the positive impression. 

Question 4. Have the authors made all data underlying the findings in their manuscript fully available? 

Reviewer #3: Yes

Reviewer #4: (No Response)

Response: Thank you for the feedback and for the positive impression. 

Question 5. Is the manuscript presented in an intelligible fashion and written in standard English?

Reviewer #3: No

Reviewer #4: Yes

Response: Thank you for your comment and do acknowledge the feedback. 

6. Review Comments to the Author

Reviewer #3: The authors have worked to respond to the comments. The paper still requires a careful read for typos and grammar. For example, in the abstract, “Proposition of facilities that

offered the service for diabetes, cardiovascular disease, chronic respiratory disease, cancer

diseases, mental illness, and chronic renal diseases was calculated to measure health service

availability” should be corrected to proportion. In the introduction “and the incidence and prevalence of NCDs are become rising”

Response: Thank you for your comment and do acknowledge the feedback. 

Question :Abstract: I was not sure what “However, in 2014 , the mean availability of

tracer items was 7% each for mental illness and chronic renal diseases, each respectively” signified-what were the results in more recent years? Same comment for discussion

Response: It is a good point raised by the reviewer. It is to mean , “ However, in 2014 survey year” , we added the word survey year so that it will suggest the data collection period where the survey has been conducted. 

Question: Methods: I remain confused about “A readiness score

of 50 signifies that, on average, half of the facilities that offered the service had each of the

requisite inputs for delivering that service. The calculated indices were compared to an agreed

cutoff point of 50%. A facility index that was below 50% was considered as `not ready' to

manage NCDs.”. This seems to imply use overall and then later in the sentence about an individual facility. There for is a score of 40% that 40% of facilities were above 50%??

Response: It is quite good point suggested by the reviewer. Actually in the method section of “ measurement of variables” , it is clearly mentioned that the readiness score was calculated with the following domains: included trained staff, guideline, equipment/supplies; medicines and commodities, and diagnostics capacity. So the cut of point of 50% will be surfaced to assess the health facility capacity in that line to differentiate the health facility readiness for managing NCDs. 

Question: Results- Statistically significant is not adequate-you need to put in the results in the text

Response: it is a good comment forwarded by the reviewer and amended in the revised file. 

Question: The number of % of facility by type seems to have changed a lot across the surveys-with many more lower level facilities early on. How did the authors adjust and if not, need to add to limitations that some change could be due to different make-up of the sample.

Response: the distribution of facilities by type, some of the heath facilities types are overrepresented and some types are underrepresented. For example, primary hospitals in Ethiopia account for only about 1% of all health facilities. Most of the country health facilities are designed under primary health care(PHC) where the majority of the health facilities were run by the heath center and health post (i.e. In actuality, about 65% of health facilities in Ethiopia are health posts). 

Question: In the discussion, please include the dates for the surveys being used in comparisons (South Africa and Uganda). Same for the data from Ethiopia

Response: It is a good comment and amended in the revised manuscript 

Question: Table 1: Would be easier to interpret if the columns included both % and n rather than having the reader calculate the number of facilities

Response: It is a good comment. Considering the numbers of health facilities of each type and national and regional levels in to account for the weighted/ adjustment , so that the distribution of facilities by type’s contribution to the total is proportionate to reflect the actual distribution or total number of health facilities in Ethiopia. Hence, the weighted percent is the one depicted in the table , that actual show the true picture of the health facilities in the country. The explanation is the percentages in the table are weighed and the number of facilities (n) in the last columns should be Unweighted. Unfortunately, we have included the weighted for SPA+ 2014. Now we have changed to Unweighted numbers (n= 873). In principle we can say/write the x %age of facilities offer diabetes service in 2014 in text but we can't put the observation (n) for every %ages in the table, It's conventional that, we need to weight estimates or percentages to make sampling adjustment for disproportional sampling and non-response rate at the facility type and regional level. This is a guiding principle for the analysis of SPA data.

Question: Table 3: Would limit p values to 2 decimal points if not significant. I also assume difference is absolute, nor relative but that should be made clear

Response: we have rounded up to two decimal points for the p-value and it is an absolute difference. 

Question: Reviewer #4: Substantial revisions have been made from the feedback of previous reviewers. I wish to however draw your retention to table 1. In the last row of totals, it is not clear what it represents. Attempt to add urban and rural located facilities gives conflicting totals from what is imputed in most of the cells. It may also be important to give a bit more details about the table as a footnote, including spelling out abbreviations/acronyms so that the table can be understandable if standing alone (this should also apply to other tables/figures). it may also be important to include the denominator number for each characteristic being measured for a better understanding of the proportions.

Response: It is a greet feedback that posed from the reviewers point of view. We will improve the design and layout as per the recommendation from the reviewer side.

---

## [Decision Letter · Decision Letter 3]

10 Jan 2024

Health System’s Availability and Readiness of Health Facilities for Chronic Non-Communicable Diseases: Evidence from the Ethiopian National Surveys

PONE-D-22-03523R3

Dear Dr. Biadgilign,

We’re pleased to inform you that your manuscript has been judged scientifically suitable for publication and will be formally accepted for publication once it meets all outstanding technical requirements.

Kind regards,

Charu C Garg, Ph.D.

Academic Editor

PLOS ONE

Additional Editor Comments (optional):

While most comments have been addressed, there are still some grammatical errors. Also some changes that are made in track change version do not reflect in the main manuscript. eg. 363 health facilities in track changes are still showing as 873 health facilities in 2014 in the main paper. Plus this number is missing in the results and sentence looks incomplete. There are several other inconsistencies like these. Request the authors to clean all these inconsistencies before the paper is published.

Reviewers' comments:

Reviewer's Responses to Questions

**Comments to the Author**

1. If the authors have adequately addressed your comments raised in a previous round of review and you feel that this manuscript is now acceptable for publication, you may indicate that here to bypass the “Comments to the Author” section, enter your conflict of interest statement in the “Confidential to Editor” section, and submit your "Accept" recommendation.

Reviewer #3: All comments have been addressed

2. Is the manuscript technically sound, and do the data support the conclusions?

Reviewer #3: (No Response)

3. Has the statistical analysis been performed appropriately and rigorously? 

Reviewer #3: (No Response)

4. Have the authors made all data underlying the findings in their manuscript fully available?

Reviewer #3: (No Response)

5. Is the manuscript presented in an intelligible fashion and written in standard English?

Reviewer #3: (No Response)

6. Review Comments to the Author

Reviewer #3: the authors have done a through review and response. It still needs a final read for grammar and clarity. T

7. PLOS authors have the option to publish the peer review history of their article (what does this mean?). If published, this will include your full peer review and any attached files.

Reviewer #3: No

---

## [Editor Report · Acceptance letter]

16 Feb 2024

PONE-D-22-03523R3 

PLOS ONE

Dear Dr. Biadgilign, 

I'm pleased to inform you that your manuscript has been deemed suitable for publication in PLOS ONE. Congratulations! Your manuscript is now being handed over to our production team.

Kind regards, 

on behalf of

Dr. Charu C Garg 

Academic Editor

PLOS ONE